# ATP Inhibits Breast Cancer Migration and Bone Metastasis through Down-Regulation of CXCR4 and Purinergic Receptor P2Y11

**DOI:** 10.3390/cancers13174293

**Published:** 2021-08-26

**Authors:** Xiaowen Liu, Manuel A. Riquelme, Yi Tian, Dezhi Zhao, Francisca M. Acosta, Sumin Gu, Jean X. Jiang

**Affiliations:** 1Department of Biochemistry and Structural Biology, University of Texas Health Science Center, 7703 Floyd Curl Drive, San Antonio, TX 78229-3900, USA; 2204110107@csu.edu.cn (X.L.); Riquelme@uthscsa.edu (M.A.R.); 2204110204@csu.edu.cn (Y.T.); zhaod1@uthscsa.edu (D.Z.); AcostaFM@uthscsa.edu (F.M.A.); gus@uthscsa.edu (S.G.); 2Department of Oncology, Xiangya Hospital, Central South University, Changsha 410008, China; 3Department of Thoracic Surgery, Second Xiangya Hospital, Central South University, Changsha 410011, China

**Keywords:** ATP, triple-negative breast cancer, CXCR4, P2Y11

## Abstract

**Simple Summary:**

The skeleton is the most frequent metastatic site for advanced breast cancer, and complications resulting from breast cancer metastasis are a leading cause of death in patients. Therefore, the discovery of new targets for the treatment of breast cancer bone metastasis is of great significance. ATP released by bone osteocytes is shown to activate purinergic signaling and inhibit the metastasis of breast cancer cells in the bone. The aim of our study was to unveil the underlying molecular mechanism of ATP and purinergic signaling in inhibiting the bone metastasis of breast cancer cells. We demonstrated that CXCR4 and P2Y11 are key factors in regulating this process, and understanding of this important mechanism will aid in identifying new targets and developing first-in-class therapeutics.

**Abstract:**

ATP released by bone osteocytes is shown to activate purinergic signaling and inhibit the metastasis of breast cancer cells into the bone. However, the underlying molecular mechanism is not well understood. Here, we demonstrate the important roles of the CXCR4 and P2Y11 purinergic receptors in mediating the inhibitory effect of ATP on breast cancer cell migration and bone metastasis. Wound-healing and transwell migration assays showed that non-hydrolysable ATP analogue, ATPγS, inhibited migration of bone-tropic human breast cancer cells in a dose-dependent manner. BzATP, an agonist for P2X7 and an inducer for P2Y11 internalization, had a similar dose-dependent inhibition on cell migration. Both ATPγS and BzATP suppressed the expression of CXCR4, a chemokine receptor known to promote breast cancer bone metastasis, and knocking down CXCR4 expression by siRNA attenuated the inhibitory effect of ATPγS on cancer cell migration. While a P2X7 antagonist A804598 had no effect on the impact of ATPγS on cell migration, antagonizing P2Y11 by NF157 ablated the effect of ATPγS. Moreover, the reduction in P2Y11 expression by siRNA decreased cancer cell migration and abolished the impact of ATPγS on cell migration and CXCR4 expression. Similar to the effect of ATPγS on cell migration, antagonizing P2Y11 inhibited bone-tropic breast cancer cell migration in a dose-dependent manner. An in vivo study using an intratibial bone metastatic model showed that ATPγS inhibited breast cancer growth in the bone. Taken together, these results suggest that ATP inhibits bone-tropic breast cancer cells by down-regulating the P2Y11 purinergic receptor and the down-regulation of CXCR4 expression.

## 1. Introduction

Breast cancer is the most common malignancy among women worldwide [1]. Although breast cancer patients’ overall survival has improved in recent decades, it is still the primary cause of cancer deaths in women. Distant metastasis is considered the primary cause of treatment failure. Bone is the most common metastatic site for breast cancer, and bone metastases cause significant complications, which greatly compromise the quality of life and shorten survival time [2,3]. In addition to its role as an intracellular energy source, similar to other cell types, extracellular ATP in cancer cells acts as a ligand that binds to purinergic receptors and modulates intracellular signaling mechanisms [4]. ATP binds to different P2 purinergic receptors and exhibits different effects, depending on the extracellular ATP concentration, the engaged P2 receptor subtypes, and the target cell types [4]. Previous studies have revealed the anti-tumor function of extracellular ATP in inhibiting the growth of some cancer cell lines, such as melanoma cells, bladder carcinoma cells, colon adenocarcinoma cells, and prostate cancer cells [5,6,7]. We have previously shown that ATP release by connexin hemichannels in osteocytes suppresses breast cancer cell migration and bone metastasis [8,9]. Moreover, we showed that, contrary to ATP, adenosine, an ATP product, has an adverse effect that promotes breast cancer migration and growth [8]. Furthermore, ATP exhibits a biphasic effect on MDA-MB-231 breast cancer cells; a lower concentration suppresses, while a high concentration promotes breast cancer cell migration. This effect is primarily caused by the instability of ATP and adenosine formation due to enhanced levels of ATPase activity exhibited by metastatic breast cancer cells [10,11]. However, how ATP regulates the underlying purinergic signaling mechanism and its downstream effectors in breast cancer bone metastasis suppression is not well understood.

P2X7 is the most studied member of the P2 receptor family. Studies have shown that the activation of P2X7 receptors has various effects on different tumor cells. The activation of P2X7 receptors can promote the proliferation and growth of human pancreatic cancer cells by increasing the phosphorylation of ERK1/2 and JNK [12]; P2X7 can also inhibit the migration of breast cancer-derived endothelial cells by activating the cAMP signaling pathway [13]. In addition to P2X7, recent studies have found that the activation of P2Y11 receptors also has a biphasic effect on tumor cells. The activation of P2Y11 receptors can change the cell cycle of prostate tumor cells from a proliferative state to a differentiated state [14] but promote the growth and migration of tumor cells in liver cancer cells [15].

CXCR4 has been shown to be involved in tumor growth and metastasis of breast cancer [16]. The binding of CXCR4 to its ligand SDF-1 (CXCL12) stimulates tumor cell proliferation and migration by activating downstream signaling transduction pathways, including the PI3K-AKT and MAPK pathways [17]. The CXCL12/CXCR4 axis plays an important role in directing CXCR4-overexpressed breast cancer cells’ metastasis to organs that express high levels of CXCL12, including bone marrow [18]. More studies suggest not only a critical role of CXCR4-CXCL12 in organ-specific metastases of breast cancer, but also in regulation of multiple levels of breast cancer progression, including proliferation, angiogenesis, and modulation of the tumor microenvironment [19,20,21]. Potential therapeutic strategies have been developed by targeting CXCR4 or its ligand, and efficacy has been shown in inhibiting primary tumor growth and metastasis of breast cancer [22,23,24]. In this study, we unveiled a new mechanism underlying ATP-modulated purinergic signaling and the CXCR4-CXCL12 pathway in the suppression of breast cancer bone metastasis.

## 2. Results

### 2.1. ATPγS and BzATP Inhibit Breast Cancer Cell Migration

We have shown previously that ATP suppresses breast cancer growth in the bone, but an ATP product, adenosine, promotes cancer growth [8]. To determine the effect of ATP on bone-tropic breast cancer cells, we used a non-hydrolysable ATP derivative, ATPγS (adenosine 5-[γ-thio] triphosphate tetralithium salt). ATPγS was used to treat MDA-BoM-1833, a bone-metastatic tropic breast cancer cell line derived from a well characterized MDA-MB-231 cell line [25,26]. The effect of ATPγS on cancer cell migration was determined by wound-healing (Figure 1A) and transwell migration (Figure 1B) assays. The results showed that ATPγS inhibited MDA-BoM-1833 cell migration in a dose-dependent manner. The results from both assays consistently showed a significant reduction in cell migration at ATPγS concentrations of 100 μM or higher. ATPγS at the same concentrations had no effect on proliferation and viability of the cells determined by WST-1 cell viability assay (Figure 1C), suggesting that reduced cell numbers did not cause the decrease in cell migration. Consistent with bone tropic MDA-BoM-1833 cells, ATPγS dose-dependently inhibited another breast cancer cell, MDA-MB-231 (Figure 1D).

ATP is known to activate purinergic receptors and consequently influence breast cancer progression and metastasis [14]. Our previous study showed that activation of the P2X7 channel is likely to be involved in the inhibitory effect of ATP on breast cancer MDA-MB-231 cells [8]. To manifest the possible involvement of the P2X7 channel in bone-tropic MDA-Bo-1833 cells, we used a P2X7 agonist, ATP analogue 2′,3′-(4-benzoyl)-benzoyl-ATP (BzATP). Similar to ATPγS, both wound-healing and transwell migration assays showed that BzATP consistently inhibited MDA-Bo-1833 cell migration in a dose-dependent manner (Figure 2A,B), and significant inhibition was observed at concentrations of 100 μM or higher. WST-1 measurement showed that BzATP had some significant effects on cell viability (Figure 2C); however, the impact of BzATP on cell migration at the same concentration was greater than cell proliferation, indicating that BzATP has a stronger impact on the inhibition of cell migration than proliferation and viability. A similar dose-dependent inhibition was also observed for MDA-MB-231 cells using both wound-healing (Figure 2D) and transwell (Figure 2E) migration analyses.

### 2.2. Mitigation of CXCR4 Expression by ATPγS and BzATP, and Attenuation of Inhibitory Effect of ATPγS on Cancer Cell Migration with CXCR4 Knockdown

To dissect the factors that play a key role in suppressing bone tropic breast cancer migration by ATP, we decided to explore CXCR4, a α-chemokine receptor for CXCL12 ligand. This receptor is reported to be actively involved in breast cancer bone metastasis by directing the CXCR4-positive cancer cells to organs that express high levels of CXCL12, such as bone marrow [18]. Moreover, CXCR4 expression is increased in MDA-BoM-1833 cells compared with parental MDA-MB-231 cells [25]. Real time quantitative RT-PCR (qRT-PCR) results showed that compared with non-treated control, ATPγS at 100 µM significantly inhibited CXCR4 mRNA level (Figure 3A). Western blots using anti-CXCR4 antibody also showed a significant decrease in CXCR4 protein expression after 4 and 8 h of treatment with ATPγS (Figure 3B). Cells were treated with various concentrations of ATPγS for 4 h and Western blots showed a dose-dependent reduction in CXCR4 protein expression (Figure 3C). To further elucidate the role of CXCR4 in MDA-BoM-1833 cells, CXCR4 gene expression was completely knocked down by siRNA (Figure 3D). Deletion of CXCR4, as with the treatment of ATPγS, showed significant inhibition of the migration of MDA-BoM-1833 cells using a transwell migration assay (Figure 3E). Interestingly, the inhibitory effect of ATPγS on the migration of MDA-BoM-1833 cells was compromised with a deficiency of CXCR4 by siRNA since the level of cell migration is similar comparing CXCR4-knocked down cells with or without ATPγS. BzATP had a similar effect on the reduction in CXCR4 expression (Figure 3F). These results suggest that downregulation of CXCR4 expression is one of the mechanisms for exerting the inhibitory effect of ATP on bone tropic cancer cell migration.

### 2.3. Inhibition of P2Y11 Attenuates Bone Tropic Cancer Cell Migration and CXCR4 Expression

BzATP, a purinergic agonist, primarily targets the P2X7 receptor. To determine P2X7 receptor involvement in mediating the role of ATPγS on MDA-BoM-1833 cell migration, we used a specific P2X7 antagonist, A804598. Transwell migration assay showed that A804598 had no effect on cell migration, and co-treatment with ATPγS did not attenuate the inhibitory effect exhibited by ATPγS (Figure 4A). This result indicated that the P2X7 receptor was unlikely to be involved. To further validate this hypothesis, we pre-treated the cell with 0.3 or 0.5 µM A804598, followed by BzATP. Similar to the observation obtained from ATPγS, combined treatment failed to compromise the inhibitory effect of BzATP on cancer cell migration (Figure 4B). These data support the notion that P2X7 is an unlikely purinergic receptor responsible for mediating the inhibitory effect of ATPγS and BzATP.

In contrast to its role on the P2X7 receptor, BzATP is reported to internalize the P2Y11 receptor [27] at similar concentrations as we used in this study. To explore the responsibility of the P2Y11 receptor, we used a P2Y11 specific antagonist, NF157. Transwell assay showed that NF157 significantly inhibited MDA-BoM-1833 cell migration to an even greater extent than ATPγS (Figure 4C). After pre-treatment with NF157, ATPγS treatment did not show any additive inhibitory effect. This result posits that the inhibitory effect of ATPγS is likely mediated through influencing the P2Y11 receptor. Similar to ATPγS and BzATP, NF157 dose-dependently inhibited the migration of MDA-BoM-1833 cells (Figure 4D).

To validate the role of P2Y11 in the effects seen with ATPγS, P2Y11 mRNA expression was knocked down by siRNA. Real time qRT-PCR showed that P2Y11 expression was significantly reduced by the specific siRNA (Figure 5A). Transwell migration assay showed that knocking down P2Y11 significantly inhibited the cell migration, and treatment with ATPγS did not show any significant, further reduction in cell migration deficient of P2Y11 (Figure 5B). The P2Y11 antagonist, NF157, inhibited breast cancer cell migration in a dose-dependent manner, and significant inhibition was observed at 10 µM (Figure 4C). Consistent with the result of P2Y11 knockdown, ATPγS, did not show any additive inhibitory effect of NF157 on the migration of the cells (Figure 4D). This result further suggested that the downregulation of the P2Y11 receptor mediates the impact of ATPγS on the migration of bone tropic breast cancer cells.

We showed that downregulation of CXCR4 expression was one of the mechanisms for the inhibitory effect of ATP on bone tropic cancer cell migration. To assess the functional relationship between P2Y11 and CXCR4, we treated MDA-BoM-1833 cells with various concentrations of the P2Y11 antagonist NF157 and showed a dose-dependent decrease in CXCR4 mRNA by real time qRT-PCR (Figure 5C). Furthermore, NF157 significantly decreased CXCR4 mRNA at a greater extent than ATPγS (Figure 5D). Pre-treatment of NF157 and followed by ATPγS did not exhibit a further additional decrease in the CXCR4 expression, suggesting that ATPγS down-regulates CXCR4 through the modulation of P2Y11.

### 2.4. ATPγS Inhibits the Growth of Breast Cancer Cells MDA-BoM-1833 in the Tibia

In order to confirm the effect of ATPγS on breast cancer metastasis in vivo, we used an in vivo model of bone metastasis by intratibial injection of breast cancer cells in the bone marrow. After implantation of MDA-BoM-1833 in the bone marrow cavity of the tibia, the mice were administrated ATPγS or PBS by intraperitoneal injection. The result showed a significant reduction in tumor growth in ATPγS-treated groups (Figure 6). Compared with the control group, breast cancer cells in the ATPγS-treated group grew more slowly in mouse tibias.

## 3. Discussion

ATP is not only an energy substance but is also released into the extracellular environment through tumor cells or host cells, and becomes a key component of the tumor microenvironment. As an extracellular signaling molecule binding to purinergic receptors, ATP plays a crucial role in the growth and development of cancer [14]. The role of ATP in the tumor microenvironment is complex and diverse; it can promote the growth of tumor cells and, on the contrary, also cause the death of tumor cells [11]. Chen et al. found the pro-invasive effect of ATP on prostate cancer through ERK1/2 and p38 signaling pathways [28]. Other studies have shown that extracellular ATP around some types of tumors leads to increased cell migration and invasion [6]. Rapaport et al. reported that exogenous ATP inhibits the growth of pancreatic cancer cells and colon cancer cells by arresting the cell cycle at the S stage [29]. Subsequent research also unveiled the anti-tumor function of extracellular ATP in different types of cancer cells, including prostate cancer cells, colon adenocarcinoma cells, melanoma cells, and bladder cancer cells [5,6,29]. Published studies have also shown that ATP’s different function on tumors depends on tumor cell type, engaged P2 receptor subtype, and extracellular ATP concentration [30]. The opposing effects of ATP on cancer could be partially explained by the diverse impacts of ATP and one of ATP’s derivatives, adenosine. Our previous studies showed the biphasic effects of adenosine nucleotides on breast cancer migration and growth; ATP inhibits, whereas adenosine promotes [8]. Additionally, adenosine signaling is known to cause immunosuppression, which supports tumor progression [31]. To explore the mechanistic role of ATP in breast cancer, in this study, we used a non-hydrolysable ATP derivative, ATPγS, which mitigated the complication related to ATP degradation due to the expression of ATPase on the surface of tumor cells [32,33]. Furthermore, this study, for the first time, revealed the important functional interplay of a purinergic receptor P2Y11 and a chemokine receptor, CXCR4, in mediating the effect of ATP in inhibiting breast cancer bone metastasis.

The MDA-BoM-1833 cell model is derived from the MD-MB-231 cells, which are generated from metastasized breast cancer in the bone. Compared with parental cells, MDA-BoM-1833 cells have a strong potency in metastasizing to the bone [25]. Here, we used the MDA-BoM-1833 cell line as a relevant model to explore the specific mechanism of ATP in breast cancer bone metastasis using in vitro wound-healing and transwell cell migration assays, and an in vivo intratibial tumor implantation model. Although both scratch wound and transwell migration are standard assays for studying cell migration, we were intrigued by the observation that 100 µM ATPγS had no effect on MDA-BoM-1833 cell migration in the scratch wound assay, but significantly reduced migration in the transwell assay. A possible explanation is that scratch wound is an assay for 2D cell motility, while transwell assay is designed to study cell motility in 3D [34]. Additionally, the result from the scratch wound assay could be confounded by the released factors from wound-damaged cells, as previously reported [35]. The metastasis of cancer cells to a distal location includes several consecutive phases, from the primary tumor site to the blood vessel infiltration, survival in the circulatory system, migration to secondary organs, adhesion, survival, and proliferation in target organs and tissues [36]. The ventricular injection metastasis model is an ideal animal model for studying cancer cell metastasis [37]; however, the incidence of bone metastasis is relatively low and inconsistent. We used intratibial injection to simulate the colonization of breast cancer cells in the bone, and this model mimics the late phase of breast cancer bone metastasis. The cancer cells are injected into the bone marrow, and cell migration is required for cancer cells to invade into the bone matrix. In our study, MDA-BoM-1833 breast cancer cells were directly injected into the bone marrow cavity of the tibia, and then intraperitoneal injection of ATPγS was performed to study the effect of ATPγS on the growth of breast cancer cells in the bone microenvironment. Consistent with the in vitro study, the tumor size of the mice in the ATPγS treatment group was significantly smaller than that in the control group. The results of this study showed that systemic administration of ATPγS inhibited the proliferation of MDA-BoM-1833 breast cancer cells in bone tissue.

ATP exerts its effect through its binding to purinergic P2 receptors (P2R) and activating downstream signaling cascades. The P2R family is subdivided into two major subgroups, P2X and P2Y, which are ligand-gated ion channel receptors and G protein-coupled receptors, respectively. P2X7 is an important member of the P2X subgroup, and recent studies have shown that P2X7 is an essential mediator in cancer invasion or metastasis [38]. High P2X7 expression is associated with increased survival rates in lung cancer [39]; however, P2X7 mediates apoptosis and inhibits the growth of human skin squamous cell carcinoma in mice [40]. Here, we observed the inhibitory effect of a P2X7 agonist, BzATP, on the migration of MDA-BoM-1833 breast cancer cells. Intriguingly, we found that a P2X7-specific antagonist A804598 compound failed to alter the effect of ATPγS on breast cancer cells, suggesting that the P2X7 receptor is unlikely to be involved. Pharmacological studies in cells transfected with the P2Y11 receptor have shown similar potency between BzATP and ATPγS, which elicits comparable Ca2+ signals and cAMP production. Moreover, studies have found that ATP or BzATP at 100 μmol/L, the concentration which showed the efficacy in our study, is able to down-regulate P2Y11 receptors via internalization [27,41]. Interestingly, when P2Y11 and P2Y1 are co-expressed in the cell, ATP, BzATP, and 2meSADP, a specific P2Y1 agonist, lead to the internalization of P2Y11 receptors [42,43]. P2Y11 is a G protein-coupled receptor, an important member of the P2Y subgroup, which is widely distributed in various human tissues and cells [44]. The role of P2Y11 in tumor progression has gradually been emerging. Studies have shown that the activation of P2Y11 in hepatocellular carcinoma cells promotes the migration of cancer cells [15]. On the other hand, there is evidence that activation of P2Y11 may influence the cell cycle, resulting in the stop of cell proliferation and initiation of cell differentiation [14]. P2Y11 is also involved in the anti-tumor process of ATP in prostate cancer DU145 cells [45]. Another study reported that over stimulation of P2Y11 with excessive extracellular ATP results in defective T cell migration [46]. Therefore, we explored the role of P2Y11 in the effect of ATPγS on breast cancer cell migration. After confirming the expression of P2Y11R and P2Y1R in MDA-BoM-1833 cells, we used a P2Y11 antagonist, NF157, and P2Y11 siRNA to assess the involvement of the P2Y11 receptor. Both P2Y11 antagonists and P2Y11 siRNA inhibited the migration of breast cancer cells and attenuated the inhibitory effect of ATPγS on breast cancer cell migration. The results of this study suggest that the downregulation or antagonizing P2Y11 receptor inhibits bone metastatic breast cell migration. ATPγS, similar to ATP, is likely to induce internalization of the P2Y11 receptor. The lack of expression or functional P2Y11 receptor in the breast cancer cells reduces breast cancer migration, indicating the intrinsic activity of this receptor, as reported in other cellular systems [46,47].

The chemokine receptor CXCR4 is a rhodopsin-like G protein-coupled receptor and selectively binds to the chemokine SDF-1 (CXCL12) [48]. The interaction between SDF-1 and CXCR4 activates various intracellular signal transduction pathways and downstream effectors, and mediates cell survival, proliferation, chemotaxis, migration, and adhesion [49]. CXCR4 is highly expressed in various tumor subtypes [50,51,52,53,54,55] and is associated with tumor cell chemotaxis, invasion, and proliferation. Another study shows that mice form much smaller tumor masses when inoculating breast cancer cells with low CXCR4 expression [49]. In addition, knockdown CXCR4 expression in breast cancer cells decreases tumor growth rate in mice [56]. CXCR4 knockout or knockdown significantly reduces cell proliferation, growth, migration, and invasion [57,58]. Potential therapeutic strategies have been developing by targeting CXCR4 or its ligand using antibodies, small peptides, or small interfering RNAs (siRNAs), and efficacy has been shown in inhibiting primary tumor growth and metastasis of breast cancer [22,23,24]. We showed that ATPγS reduced the expression of CXCR4 in MDA-BoM-1833 cells at the RNA level and protein level. Moreover, we found that knocking down CXCR4 by siRNA inhibited MDA-BoM-1833 migration and ablated the inhibitory effect of ATPγS on breast cancer cell migration. The suppression of cell migration by ATPγS was attenuated to a similar level as to only siRNA expressed cells, which suggests that CXCR4 is required for the effect of ATPγS on cancer cell migration. The data, for the first time, support the notion that downregulation of CXCR4 expression is one of the mechanisms for the inhibitory effect role of ATP in bone tropic cancer cell migration. Interestingly, we showed the inhibition of CXCR4 expression by NF157 and ATPγS at relatively high doses. However, NF157 at a lower concentration also had a significant effect on the inhibition of cancer cell migration, and this concentration has a lesser effect on CXCR4 expression. Here, we could not experimentally unveil the underlying mechanism to address this difference. One of the possible interpretations is that in addition to CXCR4, other pathway(s) might be activated/inhibited by NF157 and ATPγS, contributing to the inhibition of the migration.

As depicted in the diagram (Figure 7), our data unveil a new regulatory mechanism by which persistent extracellular ATP released by osteocyte Cx43 hemichannels [9], through its binding to purinergic receptor P2Y11R, down-regulates the CXCR4 receptor, a critical factor for breast cancer growth in the bone, leading to the inhibition of breast cancer cell migration and bone metastasis. Moreover, nonhydrolyzable ATP derivatives can be potentially adopted as potential therapeutic agents that suppress CXCR4 and, consequently, breast cancer growth in the bone.

## 4. Materials and Methods

### 4.1. Cell Lines and Cell Cultures

The human triple-negative breast carcinoma cell line MDA-MB-231 cells were originally purchased from ATCC, and bone tropic cell line MDA-BoM-1833 were provided by Dr. Yihong Wan at UT Southwest Medical Center (originally generated by Dr. Joan Massague at the Memorial Sloan Kettering Cancer Center). These two cell lines were incubated in the complete medium, McCoy’s 5A Modified Media (Gibco, Carlsbad, CA, USA) supplemented with 10% fetal bovine serum (FBS) (Hyclone, Logan, UT, USA), and 1% penicillin/streptomycin (P/S) (Sigma, St. Louis, MO, USA). Cells were cultured at 37 °C in a humidified 5% CO_2_ incubator.

### 4.2. Wound-Healing Migration and Transwell Migration Assays

For wound-healing migration assay, cells were cultured in 12-well tissue culture plates at 3 × 10^5^ cells/well for MDA-BoM-1833 cells and 4.5 × 10^5^ cells/well for MDA-MB-231 cells with complete medium containing 10% fetal bovine serum until over 95% confluency was achieved. A cell-free wound gap was created in each well using a sterile 200 µL pipetman tip, and cells were incubated with serum-free medium with or without the drug treatment after a gentle rinse with serum-free media. The initial images were captured immediately following wound creation. Cells were then cultured in a 37 °C incubator for 22 h and reimaged at the identical location. The gap area of the wound gaps for initial and final images was measured and presented as the percentage of the covered area by first subtracting the final gap area from the initial one and the sum was divided by the initial gap area by NIH ImageJ software (v1.51, NIH, Bethesda, MD, USA).

For the transwell migration assay, transwell membrane inserts (6.5 mm diameter, 8-μm pore size, 10 nm thick polycarbonate membranes, 24-well) were purchase from BD Biosciences. 4 × 10^4^ MDA-BoM-1833 or 1 × 10^5^ MDA-MB-231 cells were cultured in 200 µL serum-free medium with or without drug treatment in the upper chamber. 600 µL complete medium containing 10% fetal bovine serum with or without drug treatment was added to the lower chamber. MDA-BoM-1833 or MDA-MB-231 cells were cultured for 8–20 h at 37 °C. Cells were then fixed with methanol (Fisher Scientific, Waltham, MA, USA), and cells not migrated through the membrane were removed by a small cotton swab. Migrated cells on the lower chamber’s insert were stained by Hema 3 Stat Pack (Fisher Scientific). Five images of the cells covered evenly from each insert were captured by a microscope (Keyence, Osaka, Japan) with 10× magnification, and numbers of the migrated cells were quantified.

### 4.3. Cell Proliferation Assay

Cells were seeded in 96-well tissue culture plates at 2.4 × 10^4^ cells/well, or 4 × 10^4^ cells/well for MDA-BoM-1833 or MDA-MB-231 cells, respectively, and cultured for 24 h and were then incubated with serum-free medium with or without the drug treatment for another 21 h. 10 µL WST-1 (Water Soluble Tetrazolium salts, Roche, Basel, Switzerland) was added to cell culture and incubated for 1 h at 37 °C in a 5% CO2 incubator. Cell proliferation was measured at an absorbance wavelength of 450 nm with a Synergy HT Multi-Mode Microplate Reader (Biotek, Winooski, VT, USA).

### 4.4. Quantitative Real Time qRT-PCR

Total RNA was isolated from MDA-BoM-1833 or MDA-MB-231 cells using TriReagent (Molecular Research Center, Cincinnati, OH, USA), according to the manufacturer’s instructions. cDNA was synthesized by iScript Reverse Transcription Supermix (Bio-Rad, Hercules, CA, USA), and 250 ng total RNA was used in a 20 µL reaction solution. After 1:4 dilution, 4.8 µL cDNA was added into a 10 µL real time PCR reaction solution. Quantitative real time qRT-PCR was subsequently performed using an ABI 7900 PCR device (Life Technologies, Carlsbad, CA, USA) and SYBR Green (Life Technologies) with a three-step protocol (94 °C for 30 s, 61 °C for 30 s and 72 °C for 30 s). Relative expression levels of genes were determined by the ΔΔCt method, and glyceraldehyde 3-phosphate dehydrogenase (GPDH) was used as an internal reference. Primers for human CXCR4 and P2Y11 are as follows: CXCR4-F-5′-GGCTGTAGAGCGAGTGTTG-3′, CXCR4-R-5′-GGGTTCCTTGTTGGAGTCATAG-3′; P2Y11-F-5′-AGAAGCTGCGTGTGGCAGCGTTGGT-3′, P2Y11-R-5′-ACGGTTTAGGGGCGGCTGTGGCATT-3′.

### 4.5. Preparation of Crude Cell Membrane Extract and Western Blotting

Cells were collected with lysis buffer (5 mm Tris buffer containing 5 mm EDTA/EGTA) supplemented with protease inhibitors and ruptured by passing through a 20-gauge needle 20 times. Cell nuclei fraction was removed by centrifuging at 1000× *g* for 5 min. The supernatant was centrifuged at 45,000× *g* for 45 min. The pellet containing crude membrane extract was resuspended in lysis buffer supplemented with protease inhibitors and 1% sodium dodecyl sulfate (SDS). The concentration of membrane protein was measured by Micro BCA Protein Kit (Thermo Scientific, Rockford, IL, USA). The sample was mixed with SDS loading buffer (62.5 mm Tris, 2% SDS, 0.1% bromophenol blue, 10% glycerol and 5% β-mercaptoethanol) and incubated at room time for 30 min before loading on a 10% polyacrylamide gel and electro-transferring onto nitrocellulose membranes. The membranes were blocked with 10% fat-free milk and then incubated with anti-CXCR4 antibody (1:100 dilution) (Abcam, Cambridge, MA, USA) or anti-β-actin antibody (1:5000 dilution) (Sigma, St. Louis, MO, USA) overnight at 4 °C, and followed by probing with goat anti-rabbit IgG-conjugated IRDye^®^ 800CW or goat anti-mouse IgG-conjugated IRDye^®^ 680RD (1:15,000 dilution) (Fisher scientific, Waltham, MA, USA), respectively. The signal was detected using a Licor Odyssey Infrared Imager (Lincoln, NE, USA), and image quantification was performed by NIH ImageJ software.

### 4.6. Small Interfering RNA (siRNA) Transfection

All small interfering RNA (siRNA) including siCXCR4 (s15412), siP2Y11 (s194675), and silencer negative control siRNA are predesigned through the Silencer Select siRNA (Ambion, Austin, TX, USA). siRNA transfection was performed according to the manufacturer’s instruction (Ambion, Austin, TX, USA). Briefly, cells were cultured in a 6-well tissue culture plate for 24 h to reach about 70–90% confluency. Totals of 9 μL Lipofectamine 2000 (Invitrogen, Carlsbad, California, USA) and 3 μL 10 µM siRNA were diluted in 150 μL Opti-MEM medium, respectively. The mixture of diluted Lipofectamine 2000 and diluted siRNA at 1:1 ratio was incubated for 5 min at room temperature. A total of 100 μL Lipofectamine-siRNA mixture was added to each well, and cells were then incubated for 48 h.

### 4.7. Intratibial Injection and Bioluminescence Imaging

Six-week-old female nude mice were randomly distributed into two groups: the experimental group and the control group. MDA-BoM-1833 were implanted into mice by intratibial injection (20,000 cells/mouse). The mice in the experimental group were treated with 500 μL ATPγS (400 μmol/mL, three times/week), and mice in the control group were given an equal volume of PBS (three times/week) by intraperitoneal injection. Bioluminescence imaging system (Xenogen IVIS-Spectrum imaging system, Alameda, CA, USA) was used to detect tumor formation in mice and monitor tumor growth weekly. Our animal experiment received ethical approval from the UTHSCSA Institutional Animal Care and Use Committee (IACUC), and the corresponding ethical approval code is 20060124AR.

### 4.8. Statistical Analysis

Graphpad Prism 5 was used to perform all the statistical analyses in this article. All data in this article are presented as mean ± standard error (“X” ± SEM). Data analysis uses Student’s *t* test or one-way analysis of variance (one-way ANOVA). The asterisk indicates the statistically significant differences between groups. (*, *p* < 0.05; **, *p* < 0.01; ***, *p* < 0.001, ****, *p* < 0.0001).

## 5. Conclusions

Our data unveil a new regulatory mechanism by which persistent extracellular ATP released by osteocyte Cx43 hemichannels, through its binding to purinergic receptor P2Y11R, down-regulates the CXCR4 receptor, a critical factor for breast cancer growth in the bone. This leads to the inhibition of breast cancer cell migration and bone metastasis. Moreover, nonhydrolyzable ATP derivatives can be potentially adopted as potential therapeutic agents that suppress CXCR4 and, consequently, breast cancer growth in the bone.

## Figures and Tables

**Figure 1 cancers-13-04293-f001:**
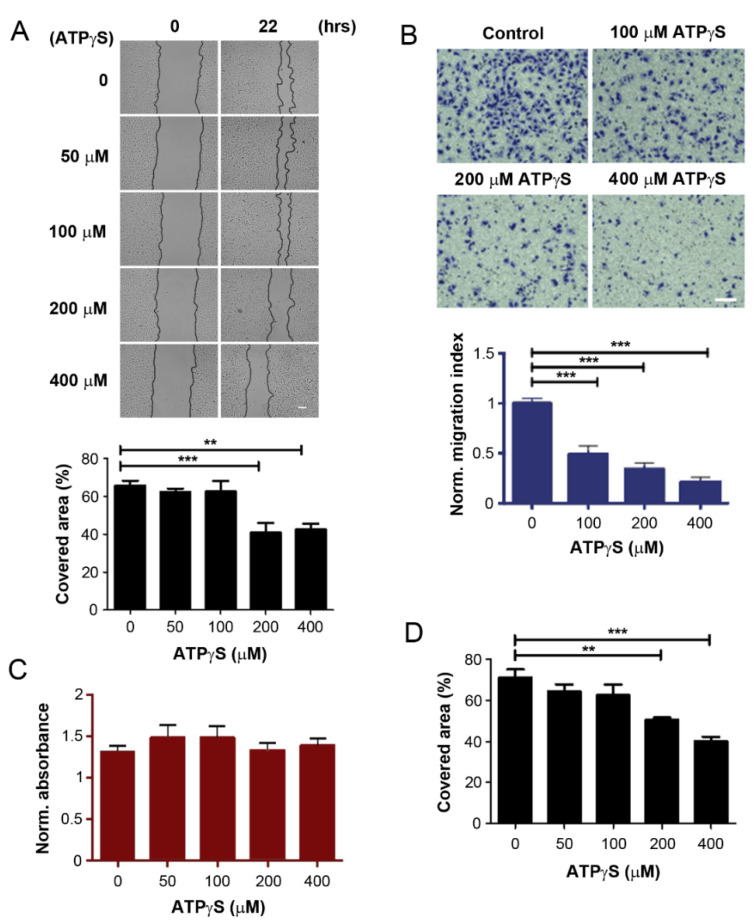
ATPγS inhibits breast cancer cell migration. (**A**) The confluent MDA-BoM-1833 human breast cancer cells were scrape-wounded and treated with various concentrations of ATPγS (upper panels). The extent of the migration was quantified by the covered migration areas using NIH ImageJ software and normalized based on the initial and final images to obtain the percentage of covered area (lower panel). Scale bar: 100 µm. (**B**) The transwell migration assay was performed by treating with various concentrations of ATPγS (upper panels) for 8 h, and MDA-BoM-1833 cells migrated across the transwell membrane were quantified and normalized to non-treated control (lower panel). Scale bar, 100 µm. (**C**) MDA-BoM-1833 cells were incubated with various concentrations of ATPγS and cell proliferation was analyzed by WST-1 assay and quantified by spectrometry at a wavelength of 450 nm (OD450). The data were normalized to those at OD600. (**D**) Wound-healing assay was performed in MDA-MB-231 cells after treatment with various concentrations of ATPγS for 22 h and quantified, representative Wound-healing assay pictures can be found at Appendix A. *n* = 3. The data are presented as mean ± SEM, ***, *p* < 0.001; **, *p* < 0.01.

**Figure 2 cancers-13-04293-f002:**
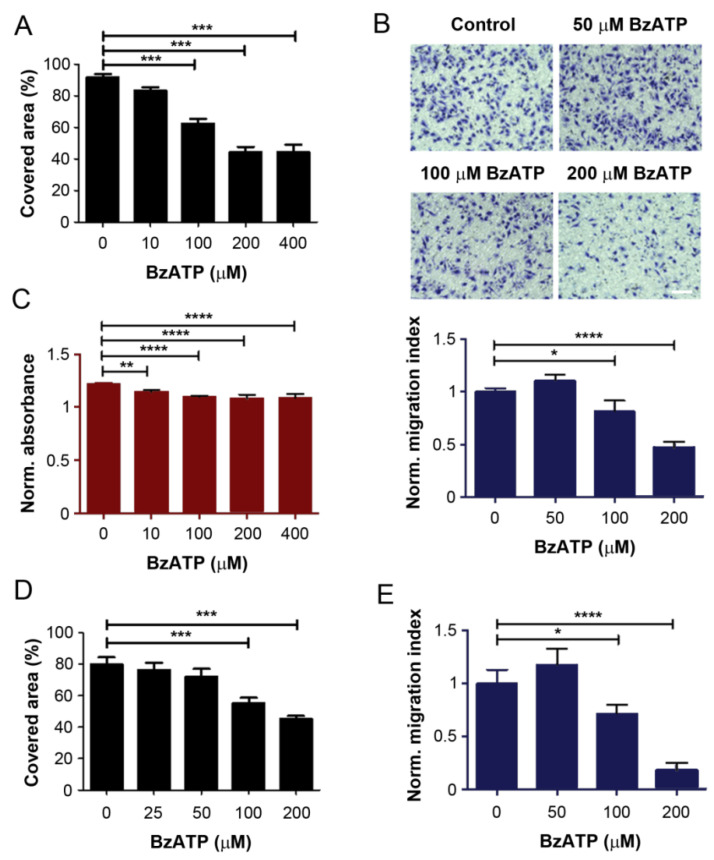
Inhibition of breast cancer migration by P2 receptor agonist, BzATP. (**A**) Wound-healing assay was performed in MDA-BoM-1833 cells with BzATP. The extent of the migration was quantified by the covered migration areas using NIH ImageJ software and normalized based on the intial and final images to obtain the percentage of covered area, representative Wound-healing assay pictures can be found at Appendix A. (**B**) The transwell migration assay was performed in MDA-BoM-1833 cells in the presence of BzATP for 8 h and the cells migrated across the transwell membrane were counted (upper panel), quantified, and normalized to non-treated control (lower panel). Scale bar, 100 µm. (**C**) Cell proliferation assay using WST-1 was performed in MDA-BoM-1833 cells with BzATP and quantified by spectrometric absorbance. at wavelength of 450 nm (OD450). The data were normalized to OD600. (**D**,**E**) MDA-MB-231 were treated with various concentrations of BzATP and scrape-wounding migration (**D**), and transwell migration (**E**) assays were performed and quantified, representative Wound-healing assay pictures can be found at Appendix A. The time periods for incubation with BzATP were 22 h and 20 h for these two assays, respectively. Transwell migration data were normalized to non-treated control. *n* = 3. The data are presented as mean ± SEM, ****, *p* < 0.0001; ***, *p* < 0.001; **, *p* < 0.01; *, *p* < 0.05.

**Figure 3 cancers-13-04293-f003:**
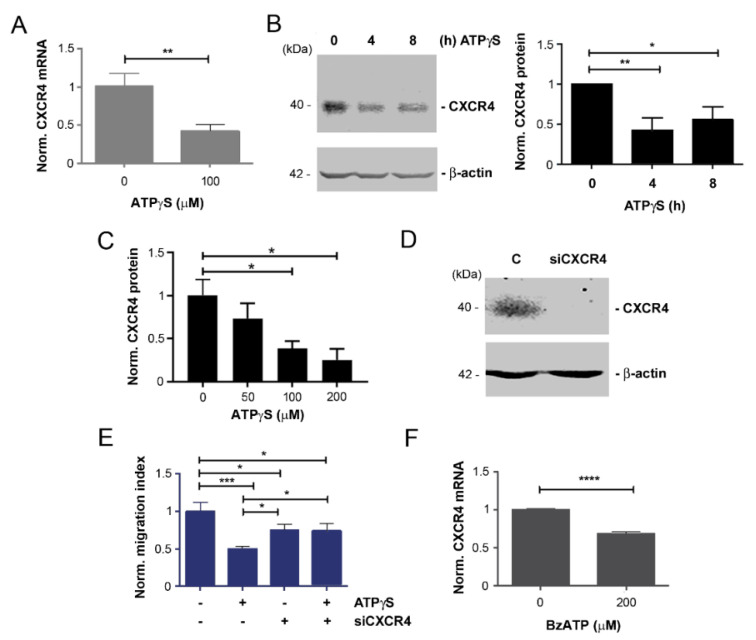
ATPγS and BzATP reduced CXCR4 expression and knockdown CXCR4 significantly attenuated the inhibitory effect of ATPγS on MDA-BoM-1833 cell migration. (**A**) MDA-BoM-1833 cells were treated with 100 µM ATPγS for 4 h and level of CXCR4 mRNA was determined by real time qRT-PCR. (**B**) MDA-BoM-1833 cells were treated with 100 µM ATPγS for 4 or 8 h and level of CXCR4 protein was determined by Western blotting with anti-CXCR4 or β-actin antibody, quantified and normalized with β-actin. Detailed information about Western Blot can be found at Appendix A. (**C**) MDA-BoM-1833 cells were treated with 50, 100, and 200 µM ATPγS for 4 h and CXCR4 protein expression was determined by Western blotting with anti-CXCR4 or β-actin antibody, quantified and normalized with β-actin. (**D**) MDA-BoM-1833 cells were transfected with CXCR4 siRNA and 48 h after transfection, Western blotting was performed with anti-CXCR4 or β-actin antibody. Detailed information about Western Blot can be found at Appendix A. (**E**) MDA-BoM-1833 cells were transfected with CXCR4 siRNAs, and 48 h later cells were treated with or without 100 µM ATPγS for 8 h. The transwell migration assay was then performed and the cells migrated across the transwell membrane were quantified and normalized with non-treated control. (**F**) MDA-BoM-1833 cells were treated with 100 µM BzATP for 4 h and level of CXCR4 mRNA was determined by qRT-PCR. *n* = 3. The data are presented as mean ± SEM, ****, *p* < 0.0001; ***, *p* < 0.001; **, *p* < 0.01; *, *p* < 0.05.

**Figure 4 cancers-13-04293-f004:**
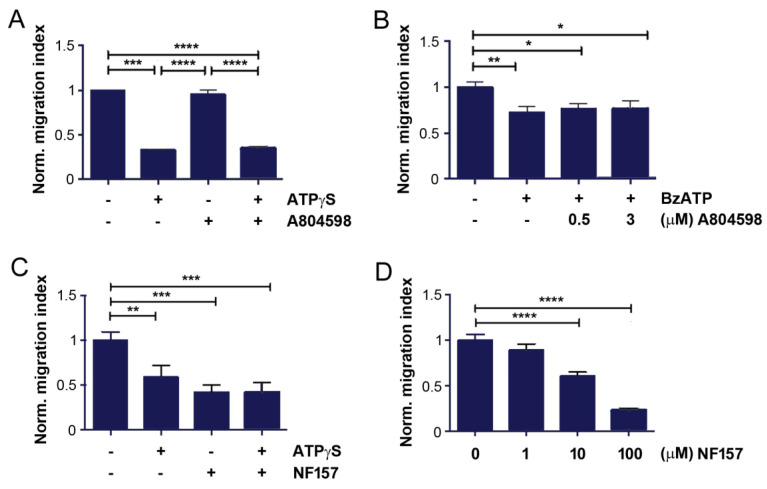
Suppression of breast cancer cell migration by inhibition of P2Y11 receptor. (**A**) MDA-BoM-1833 cells were preincubated with 100 µM A804598, P2X7 antagonist for 0.5 h and followed by treatment of 100 µM ATPγS for 12 h. Cells migrated across the transwell membrane were quantified and normalized to non-treated control. (**B**) MDA-BoM-1833 cells were preincubated with 0.3 and 0.5 µM A804598 for 0.5 h and followed by treatment of 200 µM BzATP for 12 h. Cells migrated across the transwell membrane were quantified and normalized to non-treated control. (**C**) BoM-1833 cells were preincubated with 100 µM NF157, P2Y11 antagonist for 0.5 h and followed by treatment of 100 µM ATPγS for 7.5 h. Transwell migration assay was performed and cells migrated across the transwell membrane were quantified and normalized to the non-treated control. (**D**) BoM-1833 cells were treated 1, 10, and 100 µM NF157 for 8 h and the transwell migration assay was performed. Cells migrated across the transwell membrane were quantified and normalized to non-treated control. *n* = 3. The data are presented as mean ± SEM, ****, *p* < 0.0001; ***, *p* < 0.001; **, *p* < 0.01; *, *p* < 0.05.

**Figure 5 cancers-13-04293-f005:**
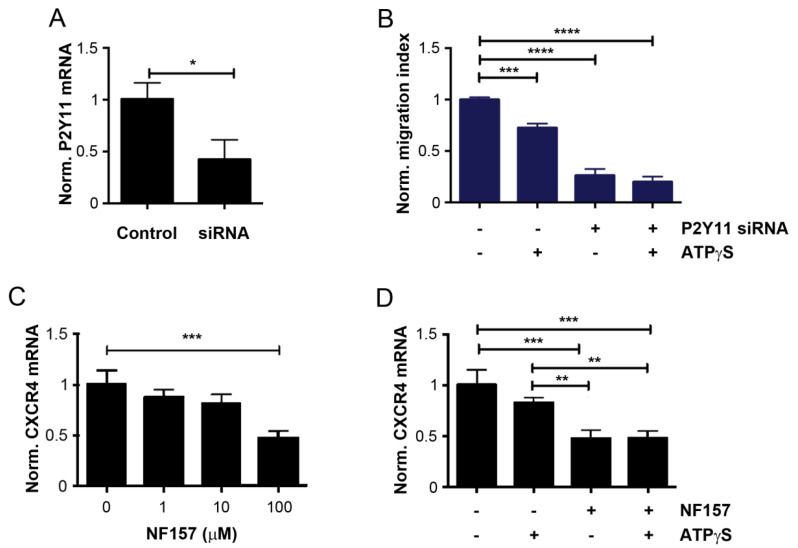
Reduction and inactivation of P2Y11 receptor decrease CXCR4 expression in breast cancer cells. (**A**) MDA-BoM-1833 cells were transfected with P2Y11 siRNA and the level of P2Y11 mRNA was quantified by qRT-PCR. (**B**) MDA-BoM-1833 cells were transfected with P2Y11 siRNA and after 48 h, cells were treated with 100 µM ATPγS for 8 h. Transwell migration assay was performed, and cells migrated across the transwell membrane were quantified and normalized to non-treated control. (**C**) MDA-BoM-1833 cells were treated with 1, 10, and 100 µM NF157 for 8 h and relative level of CXCR4 mRNA was measured using qRT-PCR. (**D**) MDA-BoM-1833 cells were preincubated with 100 µM NF157 for 0.5 h before the treatment with 100 µM ATPγS for 7.5 h. Relative level of CXCR4 mRNA was measured using qRT-PCR. The data are presented as mean ± SEM, ****, *p* < 0.0001; ***, *p* < 0.001; **, *p* < 0.01; *, *p* < 0.05.

**Figure 6 cancers-13-04293-f006:**
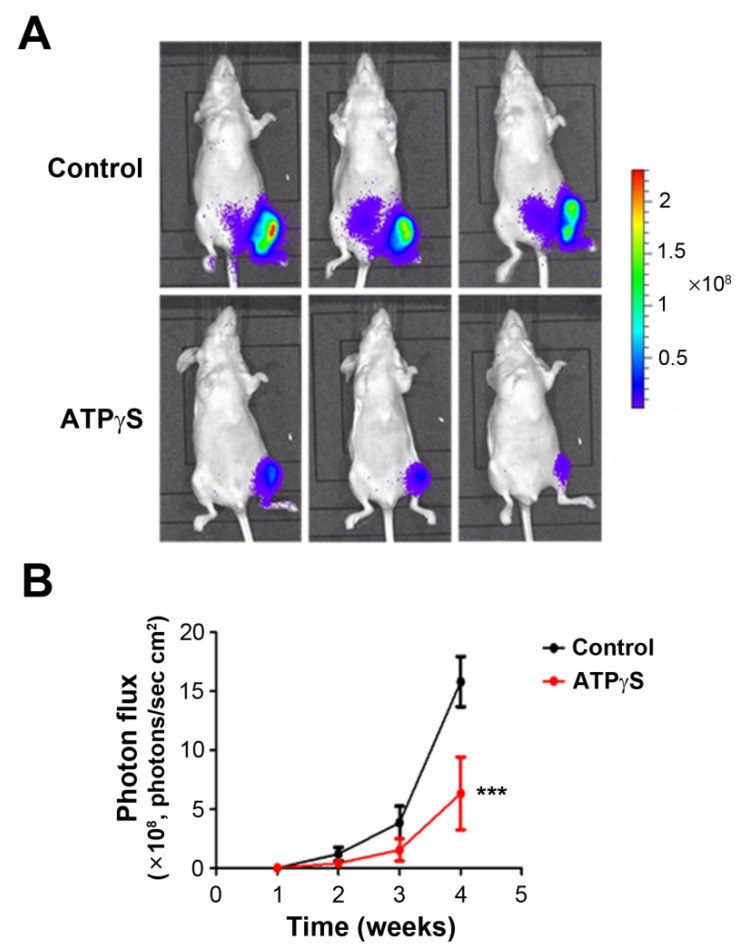
ATPγS inhibits the growth of breast cells MDA-BoM-1833 in the tibia. MDA-BoM-1833 cells were injected into the left tibia of mice with 2 × 10^4^ cells per mouse, and 500 μL ATPγS (400 μM, three times/week) or an equal volume of PBS was applied to mice by intraperitoneal injection. The bioluminescence imaging system was used to detect photon flux, an indication of tumor sizes in mice. The tumor growth was monitored weekly; the images shown were captured at the 4th week (**A**) and the photon flux value was measured by LivingImage 3.2 (**B**). *n* = 5/group, ***, *p* < 0.001.

**Figure 7 cancers-13-04293-f007:**
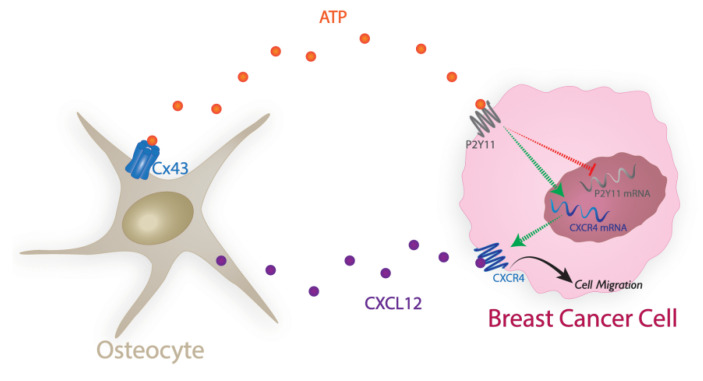
The diagram illustrating the molecular mechanism of the inhibitory effect of ATP on breast cancer bone metastasis. ATP released by osteocyte Cx43 hemichannels in the bone downregulates purinergic receptor P2Y11 through internalization in breast cancer cells. This leads to a decrease in mRNA and protein levels of CXCR4 receptor. The decreased CXCL12-CXCR4 signaling ultimately suppresses breast cancer cell migration and growth in the bone. Cx43 HC, Connexin 43 hemichannel; CXCR4, C-X-C chemokine receptor type 4; CXCL12, C-X-C Motif Chemokine Ligand 12.

## Data Availability

No new data were created or analyzed in this study. Data sharing is not applicable to this article.

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
