# Peer review of "ATP Inhibits Breast Cancer Migration and Bone Metastasis through Down-Regulation of CXCR4 and Purinergic Receptor P2Y11"

_cancers, 2021, doi:10.3390/cancers13174293_

Round 1

Reviewer 1 Report

Most of the comments and worries presented on the basis of the original version have been approached or commented in an appropriate way. The connection of the in vivo xenograft experiment remains loose and the experiment, although relevant and sound as such, fails to add information to other observations presented. A reply to a question concerning the difference of CXCR4 expression level in in cell model used (MDA-BOM-1833) remains unclear. Opposite to what is mentioned in the reply, the parental MDA MB231 cells are a characterized cell line and 1833 are a bone-seeking clone of them, also characterised in quite a detailed way (Kang et al., 2003). According to the latter, CRCR4 is expressed at an increased level in 1833 compared to parental cells.

The source of the cells used in the present study is not mentioned and therefore, it should be specified.

Author Response

Most of the comments and worries presented on the basis of the original version have been approached or commented in an appropriate way. The connection of the in vivo xenograft experiment remains loose and the experiment, although relevant and sound as such, fails to add information to other observations presented. A reply to a question concerning the difference of CXCR4 expression level in in cell model used (MDA-BOM-1833) remains unclear. Opposite to what is mentioned in the reply, the parental MDA MB231 cells are a characterized cell line and 1833 are a bone-seeking clone of them, also characterised in quite a detailed way (Kang et al., 2003). According to the latter, CRCR4 is expressed at an increased level in 1833 compared to parental cells.

Response: We appreciate the positive evaluation of our manuscript and additional comments. We believe that the reviewer referred to the differences between the responses of CXCR4 expression and cell migration in MDA-BoM-1833 cells. We showed the inhibition of CXCR4 expression by NF157 and ATPγS at relatively high doses. However, NF157 at lower concentration also had a significant effect on inhibition of cancer cell migration, and this concentration has a lesser impact on CXCR4 expression. In this paper, we could not experimentally unveil the underlying mechanism to address this difference. One of the possible interpretations is that in addition to CXCR4, other pathway(s) might be activated/inhibited by NF157 and ATPγS, contributing to the inhibition of the migration. We will test this possibility in our future research. We have included the above discussion in the text.

We agree that parental MDA-MB-231 and bone-tropic clone MDA-BoM-1833 cells are well-characterized cell lines. Based on the referred reference (Kang et al. 2003), CXCR4 expression is increased in MDA-BoM-1833 cells. We have included this in the revised text.

The source of the cells used in the present study is not mentioned and therefore, it should be specified.

Response: We have now included the source of the cells used in the Materials and Methods.

Reference:

Kang, Y., P. M. Siegel, W. Shu, M. Drobnjak, S. M. Kakonen, C. Cordón-Cardo, T. A. Guise & J. Massagué (2003) A multigenic program mediating breast cancer metastasis to bone. Cancer Cell, 3, 537-49.